# Improvement of the Thermal Stability of Polymer Bioblends by Means of Reactive Extrusion

**DOI:** 10.3390/polym15010105

**Published:** 2022-12-27

**Authors:** Félix Carrasco, Orlando Santana Pérez, Noel León Albiter, Maria Lluïsa Maspoch

**Affiliations:** 1Department of Chemical Engineering, Universitat de Girona (UdG), C/Maria Aurèlia Capmany 61, 17003 Girona, Spain; 2Centre Català del Plàstic (CCP), Universitat Politècnica de Catalunya Barcelona Tech (EEBE-UPC), ePLASCOM Research Group, Av. Eduard Maristany, 14, 08019 Barcelona, Spain

**Keywords:** thermal behavior, reaction mechanisms, reactive extrusion, PLA, PA, rheological and morphological characterization

## Abstract

Poly(lactic acid) (PLA) and biosourced polyamide (PA) bioblends, with a variable PA weight content of 10–50%, were manufactured by melt blending in order to improve the behavior of PLA against thermal degradation. The effect of reactive extrusion on the thermal performance of PLA within bioblends was analyzed. The reactive extrusion was made by means of the addition of a styrene-acrylic multi-functional-epoxide oligomeric reactive agent (SAmfE), with the commercial name of Joncryl. Four parameters were considered in order to study the thermal behavior of bioblends against thermal decomposition: the onset decomposition temperature, the shape and temperature interval of the thermal decomposition patterns, the activation energy of the thermal decomposition, and the evidence leading to the most probable mechanism. The latter was determined by means of three evidence: standardized conversion functions, y(α) master plots, and integral mean error. It was shown that reactive extrusion of PLA as well as PA incorporation to the polymer matrix of PLA were responsible for an increase in the onset decomposition temperature of 10.4 °C. The general analytical equation (GAE) was used to evaluate the kinetic parameters of the thermal degradation of PLA within bioblends for various reaction mechanisms. It was shown that the random scission of macromolecular chains is the best mechanism for both untreated and treated PLA by means of reactive extrusion. It was shown that reactive extrusion together with higher content of PA resulted in an increased protective effect against the thermal degradation of PLA as demonstrated by an increase in activation energy of 60 kJ/mol. It was found that there is a relationship between the increase in activation energy and the increase in the onset decomposition temperature when using reactive extrusion. The improvement of the thermal stability of bioblends by means of reactive extrusion was explained by an increase in the complex viscosity from 980 to 2000 Pa·s at 0.06 rad/s and from 250 to 300 Pa·s at 630 rad/s for bioblend containing 30% of PLA_REX_ and by a finer dispersion of PA within the PLA_REX_ matrix. Results from DSC were not conclusive regarding the compatibility between both phases.

## 1. Introduction

In recent decades, the plastics industry has been looking for sources of alternative raw materials to oil, with natural or renewable sources being the ones that present the greatest interest. The so-called “bio-based” polymers include natural polymers and their physical or chemical modifications (cellulose, lignin, silk, and starch) and synthetic polymers obtained from monomers derived from renewable sources. Within this last category are poly(lactic acid) or other conventional polymers whose monomer has been totally or partially obtained from renewable sources such as poly(ethylene), poly(propylene), poly(ethylene-terephthalate) and polyamides.

The biodegradability or compostability of a polymer is independent of the origin of the material since some plastics of fossil origin present this behavior. The term bioplastic or biopolymer is used both to classify materials of renewable origin, biodegradable, or those that meet both conditions. For applications with a short-life application, such as in the packaging and agriculture market, the compostability or biodegradation of the product is an important requirement. However, for long-life applications, such as in the consumer goods, automotive and electronics markets, biodegradability is of secondary importance or even a disadvantage. In these markets, more emphasis is placed on the content of raw material of renewable origin in order to reduce CO_2_ emissions associated with the product. It is precisely in this last segment of applications that our work has been focused.

Despite their great potential, bioplastics still have certain limitations and their properties are sometimes lower to those of polymers of fossil origin. In order to penetrate new markets, it is necessary to increase its performance through modifications in its macromolecular architecture [1,2,3].

Poly(lactic acid) (PLA) is the one that has aroused the greatest industrial interest. PLA is a thermoplastic linear aliphatic polyester produced by ring-opening polymerization of lactides from lactic acid. It is characterized by having mechanical properties comparable to those of polystyrene and biocompatibility. However, some of the other properties such as melt shear viscosity, melt strength, impact strength, and gas barrier properties are not good enough for certain types of applications [4]. In particular, its brittleness, due to its high rate of physical aging at room temperature, is a disadvantage for applications such as auto parts or household appliances. Although PLA has good processability in conventional transformation equipment [5], it presents high instability during processing due to hydrolytic, thermal, and oxidative degradation reactions, which lead to a decrease in its molar mass [6,7].

The attention of many research groups is focused on the modification of PLA through different strategies: the use of plasticizers, copolymerization, preparation of compounds and nanocomposites, structural modification (molecular architecture) through reactive extrusion techniques and mixing with other thermoplastics [1,2,3,7,8,9,10].

At a technical-industrial level, the addition of nanofillers still has to face several challenges related to the low level of dispersion that is usually achieved. In previous works, we verified that the best dispersion route was to apply at least two successive passes through the twin-screw extruder, which translates into a marked loss of molar mass and substantial changes in the rheological properties, such as a decrease in the viscosity of the melt. To this, it must be added the increase in the price of the product and the potential environmental and health hazards during the handling of nanofillers and waste management.

In order to alleviate the effects of the degradation produced during the preparation of these nanocomposites, it was proposed to apply reactive extrusion techniques that would allow structural modifications to be obtained in the PLA using, during a previous phase, a multiepoxidized oligomeric agent, with a functionality (epoxy groups) such that allow both extension and generation of branches. Under the processing conditions employed, these modifications promoted important rheological changes, such as the increase in shear viscosity and melt-elasticity. Likewise, it has been qualitatively confirmed that there are unreacted epoxy groups inserted in the modification that have not taken part in the extension/branching reactions [11].

Taking into account the above, as more economic and practical at an industrial level, the preparation of polymer bioblends based on the modified poly(lactic acid) and a commercial bio-based polyamide was selected as a new strategy. As occurs with fossil-based polymers, most bioplastics are partially or totally immiscible with other polymers, generating heterogeneous morphologies with low interfacial adhesion, for which compatibilization is carried out in such a way as to ensure the transmission of stresses between the phases when the system is subjected to mechanical stress. In addition to the degree of interfacial adhesion, these properties are also determined by the morphology obtained, in particular, of the final characteristics of the domains of the minor component: average size, shape, distribution, and distance between domains. This last aspect is usually controlled by the ratio and viscosity ratio between the phases at the mixing temperature [12,13].

Some authors establish that polymer blends that present a homogeneous and fine dispersion of the dispersed phase can develop a uniform distribution of the state of stresses and deformations when mechanically requested, but not in stratified systems or with coalescent phases. All this as a consequence of the high surface/volume ratio that occurs for the first case [12]. However, it has been shown that a fibrillar-type dispersion of the dispersed phase, properly compatibilized and oriented during the processing stage, can generate a reinforcing effect by increasing the fracture toughness, especially in the crack propagation stage [14]. This type of induced morphology is what has been called “in situ microfibrillated compounds” (MFCs) and have shown great potential to increase crystallization as well as the polymeric melt strength used as a matrix [15].

It has been demonstrated that some degree of compatibility between both phases was reached owing to the remaining unreacted epoxy groups in PLA_REX_. In this particular, according to isothermal crystallization studies carried out by Keridou et al. [16], PA exerts an antinucleating effect on PLA_REX_ matrix, and even the interaction between phases promotes an enhancement of lamellar twisting frequency. On the other hand, the fact of having achieved an appreciable degree of branching in modified PLA has promoted important rheological changes, which widens the window of mixing conditions to adapt it to the control of the final morphology. It allows a better ability of the generation of in situ MFC during extrusion without the need of the typical hot-stretching additional stage needed in this kind of technology. This last aspect is quiet interesting for novel manufacturing techniques such as FFF (fuse deposition modeling by pellets adding) [17].

One of the most common representatives of chain extenders is better known under its trade name Joncryl^®^. Since several years ago, Joncryl^®^ owes an ever-growing importance in academia and industry. This product largely affects rheological and thermal behavior of polymers [18]. On the other hand, Al-Itry et al. [19] demonstrated the effect of Joncryl^®^ concentration on the thermal stability of PLA/PBAT blends. For this reason, it is important to relate the molecular architecture of processed blends with its thermal stability.

The commercial significance of biopolymers is responsible for the increasing interest in the thermal behavior of PLA and its bioblends. The study of kinetics of the thermal degradation of polymers is crucial to predict their resistance to thermal decomposition. To do this, it is required to know the kinetic triplet: reaction mechanism, activation energy, and frequency factor. The knowledge of these parameters is necessary to study the thermal stability of polymer materials [20,21,22,23,24,25,26,27,28,29,30,31].

The goal of this paper was to analyze the influence of reactive extrusion of PLA on the thermal behavior of PLA/PA bioblends. Several bioblends were prepared at different PLA/PA percent compositions (90/10, 80/20, 70/30, 60/40, and 50/50), with and without reactive extrusion of PLA. The thermal stability of these bioblends was studied by means of four parameters: the onset decomposition temperature, the shape and temperature interval of the thermal decomposition patterns, the activation energy of the thermal decomposition, and the evidence leading to the most probable mechanism. The latter was determined by means of three evidence: standardized conversion functions, master plots, and integral mean error. The importance of this work consists of applying the general analytical equation to various reaction solid-state mechanisms in order to evaluate the activation energy of the thermal degradation of PLA/PA bioblends and to ascertain the beneficial effect of the reactive extrusion. Moreover, TGA data have been supported by means of rheological characterization and SEM.

## 2. Materials and Methods

### 2.1. Materials

A commercially extrusion-grade PLA (Ingeo 4032DR; was purchased from Nature-works (Arendonk, Belgium), containing 2% wt of D-lactide, with a mean molecular weight of Mn = 90 kDa and a melting temperature of 167 °C. SAmfE reactive agent (Joncryl ADR-4300FR) was provided by BASF (Ludwigshafen, Germany). This agent has an epoxy equivalent weight of 433 Da and a functionality of approximately 12. Chain extension/branching promoted in PLA results in a content of 24% wt of modified macromolecular chains, thus increasing its melt elasticity [17]. Bio-based PA10.10 was manufactured by DuPont (Midland, MI, USA), with the trade name of Zytel RS LC1000 BK385, having a melting point of 200 °C and mean molecular weights of M_n_ = 11 kDa and M_w_ = 33 kDa.

### 2.2. Bioblend Operating Procedure

Bioblends were prepared by using two different strategies: (a) PLA, as received, was used to prepare bioblends with PA; (b) PLA enhanced its thermal and mechanical properties by means of reactive extrusion with SAmfE reactive agent (weight content of 0.6%) using a corotating twin-screw extruder (Kneter 25X24D, Collin Lab and Pilot Solutions GmbH, Maintenbeth, Germany). The heating zones were set from 45 to 190 °C from feeding to die zones. The extrudate was water-cooled and pelletized. Materials were dried at 80 °C for 4 h prior to all the blending processes. In the feeding zone, a nitrogen blanket was used to minimize the degradation reactions. In the metering zone, volatiles generated during the process were removed by vacuum.

Ten PLA/PA bioblends (with 10, 20, 30, 40, and 50% of PA) were manufactured by melt mixing, using a Brabender batch mixer. This mixer was operated at a temperature of 210 °C and a speed of 50 rpm for a time of 12 min (5 bioblends without reactive extrusion of PLA and other 5 bioblends with reactive extrusion of PLA). At the end of mixing, a compression (210 °C and 4 MPa) was applied and then samples were suddenly cooled. Table 1 contains the samples used in this study.

### 2.3. Thermogravimetric Analysis

TGA data were processed on a Mettler Toledo (Columbus, OH, USA) thermogravimetric analyzer. Samples of approximately 20 mg were heated at a nominal heating rate of 10 K/min until a maximum temperature of 600 °C. A flow rate of 40 cm^3^/min of dry nitrogen was used during thermogravimetric tests.

### 2.4. Rheological Characterization

The rheological characterization of the neat polymers (PLA, PLA_REX_, and PA) and their bioblends was performed using a Small-Amplitude Oscillatory Shear Methodology (SAOS). An AR-G2 rotational rheometer (TA Instruments, New Castle, DE, USA) with a configuration of parallel plates (diameter = 25 mm) at a constant gap of 1 mm was used under a dry N_2_ atmosphere. Prior to testing, a vacuum-dried process was made overnight at 55–60 °C. Dynamic frequency sweeps were carried out at 215 °C in the angular frequency (*ω*) range 0.0628 < *ω* < 628 rad/s, under controlled deformation of 2%, to ensure that the material’s response was in the linear viscoelastic regime (LVR). The selection of the test temperature was made considering the extrusion conditions used in bioblend preparations.

### 2.5. Morphological Characterization

Bioblend morphology was assessed using a JEOL JSM-7001F scanning electron microscope (JEOL Ltd., Tokyo, Japan). The accelerating voltage select was 2 kV. Samples were collected from cryogenic fractured compression molded plates under controlled impact energy. Due to its non-conductive nature, all the samples were coated with platinum/palladium 80/20 wt %.

### 2.6. DSC Thermal Analysis

A modulated DSC Q2000 instrument was used to perform DSC (TA Instruments, New Castle, DE, USA) experiments under a dry N_2_ atmosphere. Standard Al pans were used to seal 5–6 mg of each sample, which was then subjected to a classical heating/cooling/heating procedure from 20 to 250 °C at 10 °C/min.

Figure 1 shows the experimental procedure used to obtain and characterize bioblends with and without reactive extrusion of PLA (i.e., PLA/PA and PLA_REX_/PA).

## 3. Results and Discussion

In order to analyze the behavior against thermal degradation of a material, it is important to study the following parameters: (a) The onset decomposition temperature; (b) the shape and temperature interval of the decomposition patterns; (c) the activation energy of the thermal decomposition; (d) the most probable reaction mechanism.

Table 2 shows the onset decomposition temperatures (T_5_: temperature at which 5% of material is decomposed) as a function of PA content and taking into account if PLA was previously modified by reaction extrusion or not. Independently of the reaction extrusion of PLA, onset decomposition temperatures increased when increasing PA content within the bioblend. It means that PA macromolecules dispersed into the bioblend are avoiding the depolymerization of PLA macromolecules. In addition, when PLA was previously treated by reactive extrusion, the materials reached higher onset decomposition temperatures. Therefore, reactive extrusion is also responsible for a higher thermal stability of PLA. This is a very important finding given that the material is more thermally resistant and then it will undergo less degradation during its manufacture process.

With the objective of visualizing the effect of reactive extrusion on the onset decomposition temperatures, Figure 2 shows the difference in onset decomposition temperatures with and without reactive extrusion. The increase in T_5_ was 5.0, 6.4, 7.6, 7.7, and 6.2 °C when treating PLA by reactive extrusion for bioblends containing 10, 20, 30 40, and 50% of PLA. It is clear that reaction extrusion is positive against the beginning of thermal decomposition and this benefit increased with PA content, except for the bioblend containing 50% of PA. In fact, the maximum increase was obtained for a PA content of 30–40%. As it is discussed later, the bioblend with the highest content of PA did not present a fine dispersion of PA. Therefore, the combination of reactive extrusion and higher PA content had a synergic effect against the thermal decomposition of PLA within the bioblends. The onset decomposition temperature increased 10.4 °C from bioblend containing 90% of PLA with PLA non-treated by reactive extrusion, to bioblend containing 50% of PLA with PLA treated by reactive extrusion. Hypothetically, this indicates the following: (1) Reactive extrusion of PLA is able to reduce the number of shorter macromolecular chains, which are the first to decompose; (2) PA macromolecules were adequately dispersed into the matrix, thus protecting PLA against thermal decomposition. These findings will be discussed later by means of rheological and morphological characterization.

The second parameter to be studied is the TGA shape and the temperature interval where the thermal decomposition takes place. TGA curves were transformed to curves showing the variation of conversion with temperature.

The bioblend conversion was determined from TGA data through the following equation:(1)$$\alpha =\frac{w_{o}-w}{w_{o}-w_{r}}\;$$
where *w_o_* is the initial weight, w is the weight at a given time (or temperature), and *w_r_* is the residual weight (corresponding to ash content).

Figure 3 illustrates the variation of conversion with temperature for PLA and PA neat polymers and their bioblends, when PLA was not treated by reactive extrusion, under a nominal heating rate of 10 K/min. It is clear that PA presents a significantly higher thermal resistance to thermal degradation compared to PLA. Moreover, PLA decomposition takes place with no significant degradation of PA. When 95% of PLA is decomposed (i.e., conversions of 85.5, 76.0, 66.5, 57.0, and 47.5 for bioblends containing 90, 80, 70, 60, and 50% of PLA), only 1.1–1.4% of PA is degraded for any bioblend. This finding is remarkable because it is possible to study the thermal degradation of PLA without interference of PA decomposition. In the conversion range of 85.5–47.5%, the curves are sigmoidal and therefore there is a unique value of activation energy, corresponding to the zone of PLA thermal decomposition. The same curve trends were obtained for materials where PLA was treated by reactive extrusion, but at different temperatures and with different activation energies, as it will be discussed later.

The third parameter to be studied is the activation energy for each solid-state reaction mechanism. Different conversion functions describe the kinetic mechanism of various solid-state reactions. When using the general analytical equation [32], the functions to be used for each of them are reported in Table 3, where α is the conversion, T is the temperature, A is the frequency factor, E is the activation energy, R is the gas constant, and β is the nominal heating rate. The first member of the equation takes into account the dependence of reaction rate with conversion and the second one contains the kinetic parameters (activation energy and frequency factor). By means of linear regression of the first member vs. 1/T, the slope is –E/R, from which the activation energy is calculated for each reaction mechanism.

As an example, Figure 4 shows the results obtained by means of the general analytical equation for three reaction mechanisms (F1, R1, and D1). Take into account that y-axis is the first member of the general analytical equation in linear form shown in Table 3 for each mechanism. The linear regressions for all the mechanisms presented very high regression coefficients (r^2^ > 0.99). Therefore, some additional information is needed in order to discern the best mechanism from a physical/chemical point-of-view given that all the mechanisms presented excellent fittings from a mathematical point-of-view.

The fourth parameter to be studied is the reaction mechanism responsible for the thermal degradation (remember that it was not possible to discern the best mechanism by linear regression of the general analytical equation). The most probable reaction mechanism will be determined by means of three evidences.

The first evidence is based on standardized conversion functions. Theoretical values are evaluated by taking the conversion functions shown in Table 3. Moreover, experimental values of these functions are evaluated as follows:(2)$$\frac{f\left(\alpha \right)}{f\left(\alpha_{r}\right)}=\frac{\left(d\alpha /dT\right)}{{\left(d\alpha /dT\right)}_{r}}\;exp\left[\frac{E}{R}\;\left(\frac{1}{T}-\frac{1}{T_{r}}\right)\right]$$
where $f\left(\alpha_{r}\right)$, ${\left(d\alpha /dT\right)}_{r}$, and $T_{r}$ are the conversion function, conversion derivative, and temperature at α = $\alpha_{r}$ (reference conversion), respectively. The conversion derivative and temperature at α = $\alpha_{r}$ are experimental values and E is the activation energy, previously calculated for each reaction mechanism. The reference conversions are defined as the 50% of PLA content, namely 0.45, 0.40, 0.35, 0.30, and 0.25 for bioblends containing 90, 80, 70, 60, and 50% of PLA, respectively. Figure 5 shows the variation of standardized conversion functions with PLA conversion (α_PLA_ = α/PLA content) for bioblend containing 90% of PLA non-treated by reactive extrusion when three reaction mechanisms (random scission, F2 and R1) were considered. It must be noted that the theoretical value is constant (=1) and experimental values are variable (bell shape). For this reason, R1 mechanism is absolute impossible. The evidence of standardized conversion functions is qualitative because it is based on visual divergences of these experimental and theoretical functions. It is clear that random scission and F2 presented relatively low divergences whereas R1 led to unacceptable results. When taking into account all the mechanisms, only random scission, F1, F2, R2, and R3 could be considered as potential mechanisms. Therefore, other parameters are needed in order to discern the best reaction mechanism between these five mechanisms.

The second evidence to elucidate the best reaction mechanism is based on master plots, as proposed by Criado et al. [33]:(3)$$y{\left(\alpha \right)}_{exptal}={\left(\frac{T}{T_{r}}\right)}^{2}\;\frac{\left(d\alpha /dT\right)}{{\left(d\alpha /dT\right)}_{r}}$$
(4)$$y{\left(\alpha \right)}_{theor}=\frac{f\left(\alpha \right)\;g\left(\alpha \right)}{f\left(\alpha_{r}\right)\;g\left(\alpha_{r}\right)}$$
(5)$$g\left(\alpha \right)=\int_{0}^{\alpha }\frac{d\alpha }{f\left(\alpha \right)}$$

Again, this evidence is qualitative because it is based on visual divergences between experimental and theoretical values of y(α) master plots. Mechanisms such as random scission, F1 and F2 presented again relatively low divergences. Therefore, other parameters are needed in order to discern the best reaction mechanism between three mechanisms.

The third evidence is the integral mean error (IME) of standardized conversion functions, which is defined as follows:(6)$$IME=\frac{\int_{0}^{\alpha }\left|\Delta_{f\left(\alpha \right)/f\left(\alpha_{r}\right)}\right|d\alpha }{\int_{0}^{\alpha }d\alpha }\cdot 100$$
(7)$$\Delta_{f\left(\alpha \right)/f\left(\alpha_{r}\right)}={\left(\frac{f\left(\alpha \right)}{f\left(\alpha_{r}\right)}\right)}_{exptal}-{\left(\frac{f\left(\alpha \right)}{f\left(\alpha_{r}\right)}\right)}_{theoretical}\;$$

This parameter is conclusive because it is about a quantitative procedure. For both types of bioblends (with and without reactive), the mechanism leading to lower IME values (4.6–7.2%) was that of random scission of macromolecular chains. This mechanism for neat PLA, PLA-containing blends and PLA-containing nanocomposites was already elucidated in previous works [6,34,35,36]. It must be noted that diffusion mechanisms are the least suitable given that they presented IME values of 42.0–67.3%.

Table 4 shows the values of PLA activation energy for all the bioblends (with and without reactive extrusion) for the reaction mechanism of random scission of macromolecular chains. E values increased with PA content for both untreated and treated materials by reactive extrusion. For untreated PLA, activation energy values increased from 144 to 171 kJ/mol for materials containing from 10 to 50% of PA whereas for treated materials by reactive extrusion, activation energy values increased from 154 to 180 kJ/mol. It was also shown the beneficial effect of reactive extrusion of PLA. In all cases, bioblends with treated PLA by reactive extrusion are more protected against thermal degradation. Therefore, the use of reactive extrusion in bioblends with higher content of PLA clearly protected them against thermal degradation.

In order to visualize the effect of reactive extrusion on the activation energy values, Figure 6 shows the difference in activation energy for a random scission mechanism, with and without reactive extrusion. It is clear that reaction extrusion is positive against the thermal decomposition of PLA and this benefit increased with PA content up to 30–40%. Therefore, the combination of reactive extrusion and higher PA content had a synergic effect against the thermal decomposition of PLA within the bioblends. It was found that, the activation energy increased by 60 kJ/mol from PLA (untreated by reactive extrusion) to bioblend containing 30% of PA (where PLA was treated by reactive extrusion).

It was found that there is a relationship between the main parameters studied in this work: the increase in activation energy (ΔE = E_REX_ − E_No REX_, in kJ/mol) and the increase in onset decomposition temperature (ΔT_5_ = T_5 REX_ − T_5 No REX_, in °C) when using reactive extrusion for bioblends containing 10–40%. The linear equation relating these parameters was:(8)$$\Delta E=0.8+1.9\;\Delta T_{5}\;(r^{2}=0.99)$$

This expression clearly corroborates the beneficial effect of reactive extrusion on the bioblends behavior against thermal degradation. An increase in onset decomposition temperature is accompanied by an increase in activation energy: when the onset decomposition temperature increases by 1 °C, the activation energy increases by 1.9 kJ/mol. There is an explanation to these increases in both onset thermal decomposition and activation energy: under the processing conditions employed, reactive extrusion promoted important rheological and morphological changes, which is discussed later. These results are due to the presence of extension/branching reactions, promoted by reactive extrusion, as it was also previously reported [19]. Consequently, the number of shorter chains are reduced, thus increasing the initial temperature of decomposition. On the other hand, extension/branching reactions are also responsible for an increased thermal stability of bioblends at any PLA conversion. Moreover, it was demonstrated that some degree of compatibility between both phases is reached owing to the remaining unreacted epoxy groups. As a result, PA macromolecules exert an antinucleating effect on PLA matrix [17].

The improvement in the thermal stability can be related to the molecular architecture of bioblends, through its rheological and morphological characterization. Figure 7a shows the complex viscosity of neat polymers (PLA, PLA_REX_, and PA) as a function of angular frequency. These results show a clear shear-thinning behavior under the test conditions used. The only one showing the terminal regime (Newtonian plateau) was PLA without reactive extrusion. In the case of PLA_REX_, an increase in complex viscosity was observed throughout the range of ω analyzed together with a decrease in the angular frequency of transition (ω_T_). Both aspects would indicate that the reactive extrusion induced an increase in molar mass (M_w_) and polydispersity, as well as the generation of sparsely long branching of macromolecular chains, as previously reported [37].

Within the range of shear, strain rates recorded in the extrusion process during mixing (equivalent to 100 to 300 rad/s), a similarity in complex viscosity between PA and PLA_REX_ phases was reached when PLA was treated by reactive extrusion. This fact would indicate that the use of PLA_REX_ as the major phase would potentially favor a stable break-up process for the minor phase (PA), thus promoting its greater dispersion, compared to the situation when the unmodified PLA was used.

When analyzing flow curves (η* vs. ω) of bioblends, it was observed in all cases that they lay below the one predicted by the logarithmic additive law of mixtures of viscosity. Moreover, bioblends based on unmodified PLA presented flow curves that were located between those of the neat polymers, while those based on PLA_REX_ were all located below both neat polymers. As an example, Figure 7b shows the results of complex viscosity for the bioblend containing 70% by weight of PLA, with and without reactive extrusion of PLA. Within all the range of angular frequency, the bioblend based on PLA_REX_ showed higher complex viscosities (i.e., increase from 950 to 1970 Pa·s when PLA was treated by reactive extrusion). It can be related to a higher branching/extension effect on macromolecular chains, thus improving its thermal stability (i.e., higher onset decomposition temperatures and higher activation energies).

It was reported that the weighted relaxation time spectra of starting polymers showed an increase in the distribution of relaxation times when PLA was previously treated by reactive extrusion. This fact indicates an increased number of interactions and entanglements per chains that decrease the molecular mobility [37,38].

In a previous work, García-Masabet et al. [17] reported that a very marked bimodal character was observed in these spectra, with maximum distributions located in times very close to those of the neat polymers. This fact indicates a marked biphasic nature of bioblends, with weak interaction between them. The situation changed when PLA_REX_ was used as a matrix, where the loss of said bimodality was observed, with maxima displaced in relaxation time. This could indicate some degree of interaction between both phases, perhaps promoted by the possible reactions that may arise from unreacted epoxy groups present in PLA_REX_ and PA phases.

The existence of some degree of interaction between both phases can be corroborated by analyzing the results of the tensile mechanical behavior of bioblends, which have been previously reported [13]. In the case of PLA-based bioblends, it was shown that, for low PA contents (10, 20%, and 30% by weight of PA, with a drop-shaped morphology), the values of strain at break were at the lower limit of the additive mixing rule (AMR) (limit associated with weak interaction between phases). On the opposite, for the same range of compositions of bioblends based on PLA_REX_, the values were above the prediction of the AMR in its upper limit (limit associated with good adhesion between phases). From 40% by weight of PA, for both types of bioblends, the values were above the upper limit, always being higher for those based on PLA_REX_. In this range of compositions, the greater interface between phases must be considered, so the effect cannot be attributed exclusively to a higher degree of adhesion between phases, but the factor of co-continuity or morphological transition of the PA phase also comes into play.

Figure 8 shows SEM micrographs of the cryogenically fractured surfaces of bioblends at various compositions. It was revealed a two-phase structure, independently of the composition considered. PA droplets embedded in a continuous PLA phase matrix confirm the immiscible nature of these bioblends. For bioblends with unmodified PLA, an increase in PA content induces a growth in the average size of the droplets, as shown in Figure 8a,c,e. The use of PLA_REX_ matrix induces a finer dispersion of PA phase and with no clear dependence of the size on composition, as shown in Figure 8b,d,f. This situation is expected considering the shear viscosity ratio close to one that neat polymers present at the shearing rate used in the extrusion process. Moreover, Cailloux et al. [13] found an apparently better interfacial adhesion since the PA dispersed phase was covered with the matrix.

For unmodified PLA-based bioblends, it was clearly seen a cavitation or debonding between phases. The local triaxial stress state, induced during the impact loading, is released by a debonding mechanism, thus indicating a poor interfacial adhesion between both phases. On the contrary, in PLA_REX_-based bioblends, this mechanism was not fully observed. This fact could indicate that a better interaction between phases was generated as a consequence of two combining factors: reduction of dispersed phase (increase in the surface to volume ratio of the interface) and a possible reaction between unreacted epoxide groups of the reagent coupled in PLA_REX_ with PA.

As reported by Carrasco et al. [6], the variation of microstructure of PLA processed by injection or extrusion/injection followed or not by annealing. By means of GPC, they reported a decrease in molecular weight and an increase in polydispersity index when processing PLA. In fact, degradation parameter k = M_n_ raw material/M_n_ extruded/injected = 1.64 clearly indicated an important depolymerization of PLA after processing. On the other hand, by means of H NMR they determined that there was no variation in chemical shifts of CH_3_ and CH when processing PLA, but there was an increase in CH_3_/CH relative area, thus being attributed to a variation of the molecular environment of CH_3_ and CH. In fact, CH3 has not the same chemical environment in small molecular (resulting from the degradation during PLA processing) than in bigger ones. It means that new smaller molecules were created as a consequence of depolymerization reactions. On the other hand, they have demonstrated that the onset temperature of decomposition decreased with a decrease in mean molecular weight (because of the presence of smaller molecules, which are more easily volatilized). In addition, the onset temperature of decomposition decreased with an increase in polydispersity index. Moreover, they determined by FTIR and X-ray powder diffraction the loss of crystallinity after PLA processing.

The Gibbs energy of mixing of a binary polymer blend can be calculated accounting for the combinational entropy of polymer chains and the segmental interactions. This thermodynamic variable takes into account the degree of polymerization of both polymers, the volume fraction of the two polymers and the degree of interaction, the latter reflecting the compatibility of the system, which governs the phase behavior [39]. Chen et al. [40] reported the thermodynamics of compatibility in polymer blends. These authors stated that differences in short-chain branching produce enthalpic contributions, whereas long-chain branching results in an excess entropy of mixing. Both effects increase the magnitude of the Flory–Huggins χ parameter and either induce phase separation between linear and branched polymers. It is well-known that reactive extrusion promotes extension/branching reactions and consequently the interaction parameter χ should be modified. Changes in volume fraction of each polymer (i.e., PLA and PA content) as well as variation in χ values (i.e., effect of reactive extrusion of PLA) could be responsible for a different behavior against thermal degradation, as it was demonstrated in this work (i.e., variation in onset decomposition temperature and activation energy).

It is known that DSC can be a way to assess miscibility/compatibility between phases. In fact, DSC studies were carried out to establish the possible miscibility between the phases, or failing that, the degree of interaction between them. As an example, Figure 9 shows the controlled cooling scans at 10 °C/min. The glass transition regions of the neat polymers overlap. Even the jump in specific heat for PA was attenuated, which makes it practically impossible to determine, based on the signals, any type of effect as a consequence of such interaction. On the other hand, alternatively, DMA tests were performed under conditions of linear viscoelasticity at 1 Hz and heating scan of 2 °C/min. When evaluating the specific losses (Tan δ), it was not possible to obtain anything conclusive in this regard given that the signal corresponding to the PA phase was overlapped and in the bioblends there was a high uncertainty of its contribution since the signal that could be awarded showed dependence (in terms of height) with the existing composition.

## 4. Conclusions

PLA/PA bioblends (with a predominantly biosourced PA10.10) containing 90–50% of PLA were manufactured by melt blending. The influence of reactive extrusion on the thermal behavior of bioblends was analyzed by means of four parameters, such as the onset decomposition temperature, the shape and temperature interval of the decomposition patterns, the activation energy of the thermal decomposition, and the most probable reaction mechanism.

The onset decomposition temperature (T_5_) increased from 311 to 315 °C when the PA content increased from 10 to 50%, with PLA not previously treated by reactive extrusion, and the onset decomposition temperature (T_5_) increased from 316 to 321 °C when PLA was previously treated by reactive extrusion. These data clearly confirm the beneficial effect of reactive extrusion against thermal decomposition. The onset decomposition temperature increased by 10.4 °C when using the highest content of PA and with PLA treated by reactive extrusion.

The evaluation of the kinetic parameters of the thermally activated decomposition of PLA was carried out by using the general analytical equation. We tested several solid-state reaction mechanisms to assess the best kinetic model. Three different evidence were used to reach the most probable mechanism: the standardized conversion functions, y(α) master plots, and integral mean error indexes. We demonstrated that the more accurate mechanism was the random scission of the macromolecular chains for both untreated and treated PLA by reactive extrusion. This mechanism led to integral mean errors of 2–9% for all the bioblends. Activation energy values were 144–171 kJ/mol for the bioblends where PLA was not treated by reactive extrusion and 154–180 kJ/mol for the bioblends where PLA was treated by reactive extrusion. These findings clearly indicate that a higher content of PA and the treatment by reactive extrusion were key effects against the thermal degradation of PLA. The activation energy increased from 144 kJ/mol to 204 kJ/mol from PLA non-treated by reactive extrusion to bioblend containing 30% of PA with PLA treated by reactive extrusion (i.e., increase of 42%). This finding confirms the protective effect against thermal decomposition when using reactive extrusion of PLA and increasing content of PA up to 30–40%.

It was demonstrated that there was a linear relationship between the increase in activation energy and the increase in onset decomposition temperature when using reactive extrusion: an increase of 1 °C in the onset decomposition temperature led to an increase of 1.9 kJ/mol in the activation energy.

By means of rheological and morphological characterization, it was possible to relate a higher thermal stability of bioblends when treating PLA with reactive extrusion (higher complex viscosity, increase of entanglements, some degree of interaction between phases, finer dispersion of PA phase, and more extension/branching reactions). The glass transition region of the neat polymers overlapped and there was no conclusive evidence of interaction between both phases by DSC.

## Figures and Tables

**Figure 1 polymers-15-00105-f001:**
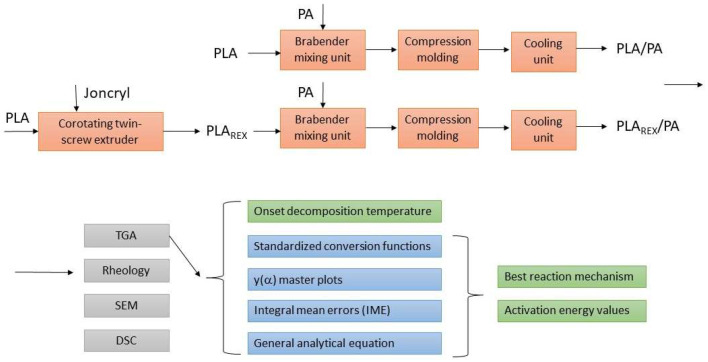
Experimental procedure used to obtain and characterize PLA/PA and PLA_REX_/PA bioblends.

**Figure 2 polymers-15-00105-f002:**
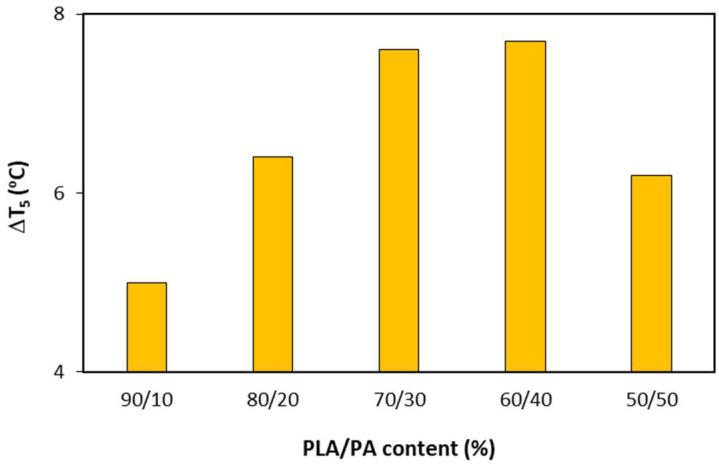
Increase in onset decomposition temperatures (ΔT_5_ = T_5 REX_ − T_5 No REX_) when using reactive extrusion for bioblends containing 90–50% of PLA.

**Figure 3 polymers-15-00105-f003:**
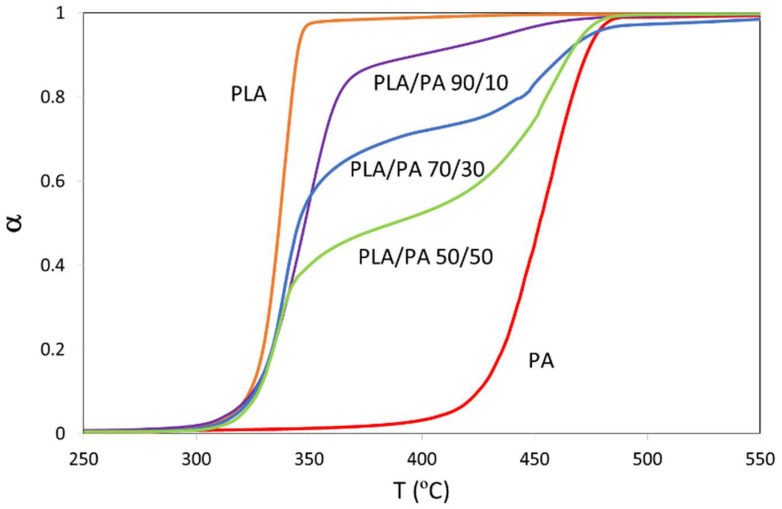
Variation of conversion with temperature for PLA and PA neat polymers, and the bioblends containing 90, 70, and 50% of PLA, without previous reactive extrusion of PLA, with a nominal heating rate of 10 K/min.

**Figure 4 polymers-15-00105-f004:**
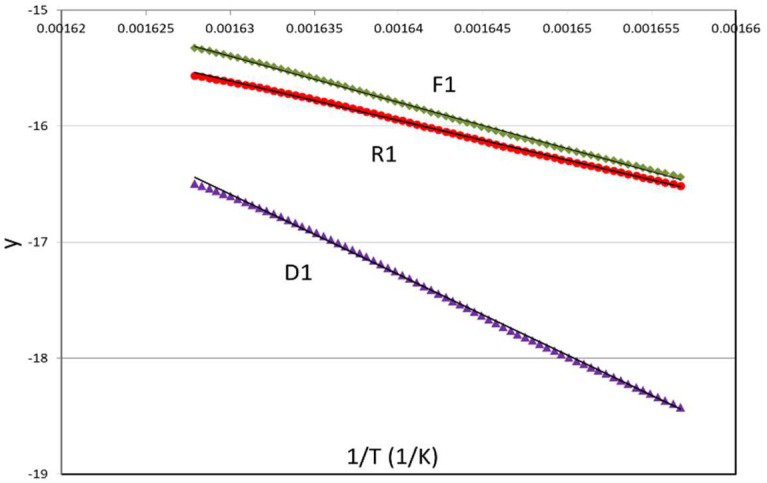
Application of the general analytical equation by linear regression for the thermal decomposition of the bioblend containing 70% of PLA treated by reactive extrusion (PLA_REX_/PA 70/30), with a nominal heating rate of 10 K/min. Note: y axis is the first member of the general analytical equation in linear form shown in Table 3 for each mechanism.

**Figure 5 polymers-15-00105-f005:**
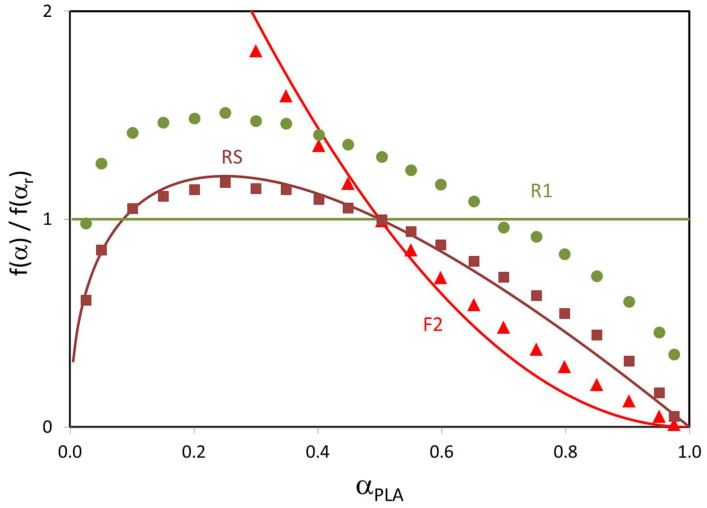
Variation of standardized conversion functions with PLA conversion for bioblend containing 90% of PLA non-treated by reactive extrusion (PLA/PA 90/10) when three reaction mechanisms were considered). Symbols are experimental values and lines are theoretical values.

**Figure 6 polymers-15-00105-f006:**
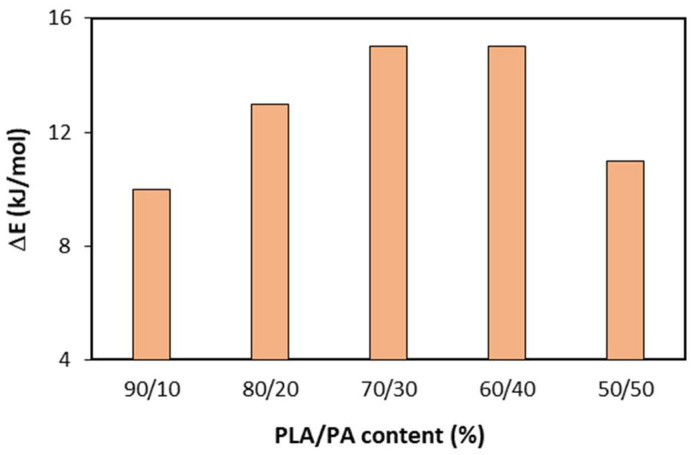
Increase in activation energy values (ΔE = E_REX_ − E_No REX_) when using reactive extrusion for bioblends containing 90–50% of PLA_REX_.

**Figure 7 polymers-15-00105-f007:**
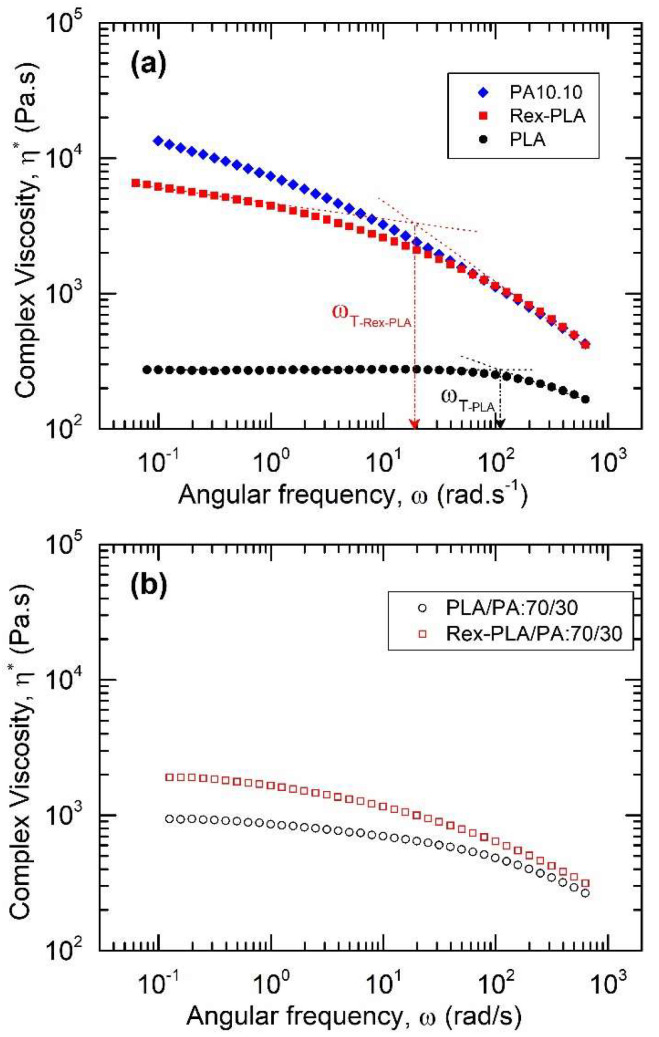
Flow curves (η* vs. ω) of neat polymers (**a**) and bioblend prepared with 70% of PLA (**b**). ω_T_: estimated angular frequency where Newtonian-shear thinning behavior is observed. T = 215 °C.

**Figure 8 polymers-15-00105-f008:**
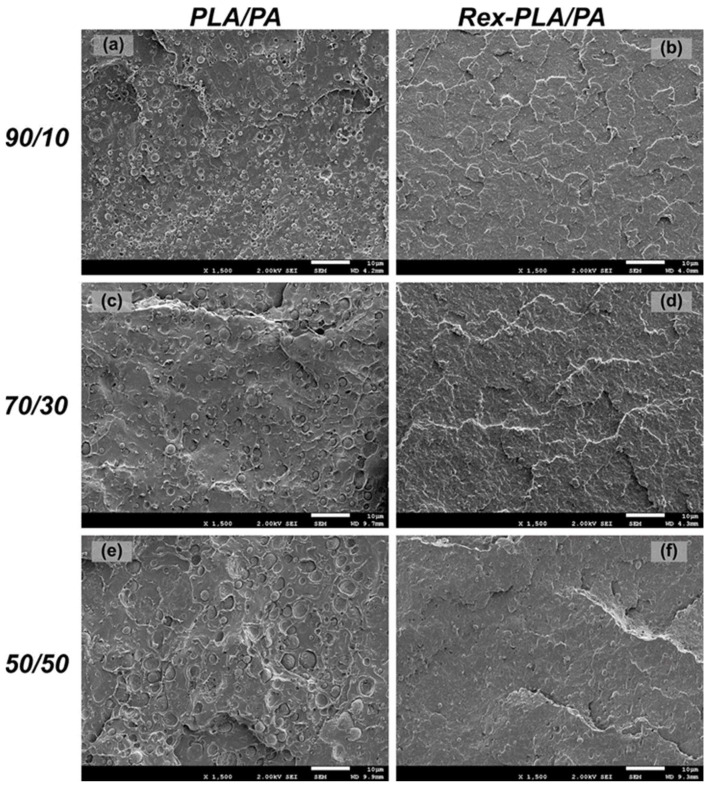
SEM micrographs (×1500) of the cryogenic fractured surfaces of bioblends at various compositions, with and without reactive extrusion. Scale bars = 10 μm. Images (**a**,**c**,**e**) correspond to bioblends without reactive extrusion of PLA for 10, 30 and 50% of PA, respectively. Images (**b**,**d**,**f**) correspond to bioblends with reactive extrusion of PLA for 10, 30 and 50% of PA, respectively.

**Figure 9 polymers-15-00105-f009:**
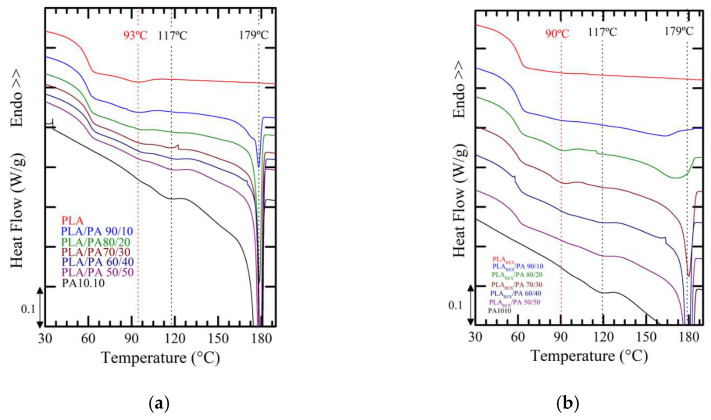
DSC cooling scans at 10 °C/min of (**a**) PLA/PA bioblends and (**b**) PLAREX/PA bioblends.

**Table 1 polymers-15-00105-t001:** Compositions of materials with or without reactive extrusion of PLA.

Sample Name	wt % of PLA	wt % of PA	PLA Reactive Extrusion?
PLA	100	0	No
PLA/PA 90/10	90	10	No
PLA/PA 80/20	80	20	No
PLA/PA 70/30	70	30	No
PLA/PA 60/40	60	40	No
PLA/PA 50/50	50	50	No
PLA_REX_	100	0	Yes
PLA_REX_/PA 90/10	90	10	Yes
PLA_REX_/PA 80/20	80	20	Yes
PLA_REX_/PA 70/30	70	30	Yes
PLA_REX_/PA 60/40	60	40	Yes
PLA_REX_/PA 50/50	50	50	Yes

**Table 2 polymers-15-00105-t002:** Onset temperatures of the thermally activated degradation of PLA/PA bioblends, with and without reactive extrusion of PLA.

	T_5_ (°C)				
	90/10	80/20	70/30	60/40	50/50
**No REX**	310.8	311.0	311.2	312.8	315.0
**REX**	315.8	317.4	318.8	320.5	321.2

**Table 3 polymers-15-00105-t003:** Kinetic equations in linear form for the solid-state mechanisms used in this study.

Mechanism	f(α)	General Analytical Equation in Linear Form
Random scission	$2\;\left(\alpha^{1/2}-\alpha \right)$	$ln\left[\beta \;\frac{-ln\left(1-\alpha^{1/2}\right)}{T^{2}\left(1-\frac{2RT}{E}\right)}\right]=ln\frac{AR}{E}-\frac{E}{R}\frac{1}{\;T}$
R1	1	$ln\left[\beta \;\frac{\alpha }{\;T^{2}\left(1-\frac{2RT}{E}\right)}\right]=ln\frac{AR}{E}-\frac{E}{R}\frac{1}{\;T}$
R2	$2\;{\left(1-\alpha \right)}^{1/2}$	$ln\left[\beta \;\frac{1-{\left(1-\alpha \right)}^{1/2}\;}{T^{2}\left(1-\frac{2RT}{E}\right)}\right]=ln\frac{AR}{E}-\frac{E}{R}\;\frac{1}{\;T}$
R3	$3\;{\left(1-\alpha \right)}^{2/3}$	$ln\left[\beta \;\frac{1-{\left(1-\alpha \right)}^{1/3}}{T^{2}\left(1-\frac{2RT}{E}\right)}\right]=ln\frac{AR}{E}-\frac{E}{R}\frac{1}{\;T}$
F1	(1−α)	$ln\left[\beta \;\frac{-ln\left(1-\alpha \right)}{T^{2}\left(1-\frac{2RT}{E}\right)}\right]=ln\frac{AR}{E}-\frac{E}{R}\frac{1}{\;T}$
F2	${\left(1-\alpha \right)}^{2}$	$ln\left[\beta \;\frac{\left(\frac{\alpha }{1-\alpha }\right)}{T^{2}\left(1-\frac{2RT}{E}\right)}\right]=ln\frac{AR}{E}-\frac{E}{R}\frac{1}{\;T}$
F3	${\left(1-\alpha \right)}^{3}$	$ln\left[\beta \;\frac{{\left(1-\alpha \right)}^{-2}-1}{2\;T^{2}\left(1-\frac{2RT}{E}\right)}\right]=ln\frac{AR}{E}-\frac{E}{R}\frac{1}{\;T}$
D1	$\frac{1}{2\alpha }$	$ln\left[\beta \;\frac{\alpha^{2}}{T^{2}\left(1-\frac{2RT}{E}\right)}\right]=ln\frac{AR}{E}-\frac{E}{R}\frac{1}{\;T}$
D2	$\frac{1}{-ln\left(1-\alpha \right)}$	$ln\left[\beta \;\frac{\left(1-\alpha \right)\;ln\left(1-\alpha \right)+\alpha }{T^{2}\left(1-\frac{2RT}{E}\right)}\right]=ln\frac{AR}{E}-\frac{E}{R}\frac{1}{\;T}$
D3	$\frac{3\;{\left(1-\alpha \right)}^{2/3}\;}{2\;\left[1-{\left(1-\alpha \right)}^{1/3}\right]}$	$ln\left[\beta \;\frac{1+{\left(1-\alpha \right)}^{2/3}-2\;{\left(1-\alpha \right)}^{1/3}}{T^{2}\left(1-\frac{2RT}{E}\right)}\right]=ln\frac{AR}{E}-\frac{E}{R}\frac{1}{\;T}$

**Table 4 polymers-15-00105-t004:** Activation energy values of PLA thermal decomposition, with and without reactive extrusion for the random scission mechanism.

	E (kJ/mol)				
	90/10	80/20	70/30	60/40	50/50
**No REX**	144 ± 2	157 ± 1	189 ± 2	190 ± 1	171 ± 1
**REX**	154 ± 2	170 ± 2	204 ± 3	205 ± 3	180 ± 2

## Data Availability

Not applicable.

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
