# Peer review of "Improvement of the Thermal Stability of Polymer Bioblends by Means of Reactive Extrusion"

_polymers, 2022, doi:10.3390/polym15010105_

Round 1

Reviewer 1 Report (Previous Reviewer 3)

The article has significant improvement and satisfy the requirement for the publication after necessary data supplied.

Author Response

REPLY TO REVIEWER COMMENTS

 REVIEWER 1

The article has significant improvement and satisfy the requirement for the publication after necessary data supplied.

Thank you very much for your kind comment.

Reviewer 2 Report (New Reviewer)

Reactive extrusion is quite an interesting method of preparing new materials with unique properties. In the case of PLA, this is a very attractive mixing process when the goal is to increase thermal resistance. From the researcher's point of view, the manuscript is very interesting.

1.       The topic of the work is consistent with its content.

2.       The literature cited is up-to-date and well-chosen, its quantity is sufficient and gives a good background for the work.

3.       The adopted research methodology is correct, the proposed research methods are well selected.

4.       My question is why do we mix PLA with PA. PLA belongs to biodegradable polymers and we care about maintaining this property of the material. The addition of PA complicates this matter. I would like some clarification on this.

5.       What is the temperature profile on the extruder for each mixture. PLA has a much lower processing temperature than PA.

6.       I would suggest supplementing the article with the study of mechanical properties, if possible.

7.       The conclusions are correctly formulated, it would be worth supplementing the work with the aspect of practical application of such mixtures.

Author Response

REPLY TO REVIEWER COMMENTS

REVIEWER 2

Reactive extrusion is quite an interesting method of preparing new materials with unique properties. In the case of PLA, this is a very attractive mixing process when the goal is to increase thermal resistance. From the researcher's point of view, the manuscript is very interesting.

Thank you very much to find that there is a very attractive mixing process in order to increase the thermal resistance of bioblends.

  1. The topic of the work is consistent with its content.

Thank you very much for your comment.

  1. The literature cited is up-to-date and well-chosen, its quantity is sufficient and gives a good background for the work.

Thank you very much for your comment.

  1. The adopted research methodology is correct, the proposed research methods are well selected.

Thank you very much for your comment.

  1. My question is why do we mix PLA with PA. PLA belongs to biodegradable polymers and we care about maintaining this property of the material. The addition of PA complicates this matter. I would like some clarification on this.

PLA has been viewed as a good alternative to commercial polymers for commodity applications (i.e. packaging). Unfortunately, its advanced brittleness at room temperature has been its major disadvantage for its implementation into high-value and durable applications (i.e. electronic or automotive). In those kind of applications, biodegradability is of secondary importance or even a disadvantage. More emphasis is placed on the content of raw material of renewable origin in order to reduce CO2 emissions associated with the product. It is precisely in this last segment of applications that our work has been focused: Preparation and characterization of PLA/bio-based PA bioblends.

This information is already indicated in the Introduction section:

“The biodegradability or compostability of a polymer is independent of the origin of the material, since some plastics of fossil origin present this behavior. The term bioplastic or biopolymer is used both to classify materials of renewable origin, biodegradable or those that meet both conditions. For applications with a short-life application, such as in the packaging and agriculture market, the compostability or biodegradation of the product is an important requirement. However, for long-life applications, such as in the consumer goods, automotive and electronics markets, biodegradability is of secondary importance or even a disadvantage. In these markets, more emphasis is placed on the content of raw material of renewable origin in order to reduce CO2 emissions associated with the product. It is precisely in this last segment of applications that our work has been focused.”

  1. What is the temperature profile on the extruder for each mixture? PLA has a much lower processing temperature than PA.

As indicated in the Materials and Methods section (Bioblend operating procedure), the bioblends used in this study were prepared in a Brabender batch mixer, operated at 210 °C and 50 rpm for 12 min. For the case of BioPA 10.10, this temperature is enough to process it and for PLA is a very common processing temperature.

On the other hand, PLA modified by reactive extrusion was produced (before bioblending) using a corotating twin-screw extruder, where heating zones were set from 45 to 190 ºC. This information is also indicated in the Materials and Methods section.

  1. I would suggest supplementing the article with the study of mechanical properties, if possible.

Mechanical properties of PLA/PA10.10 were previously studied by us and published in Express Polymer Letters [13]. Please take into account that the aim of the present study was to evaluate the thermal resistance of bioblends against thermal decomposition.

  1. The conclusions are correctly formulated, it would be worth supplementing the work with the aspect of practical application of such mixtures.

       Thank you very much for finding that conclusions are correctly formulated. Practical applications of bioplastics have been indicated in the Materials and Methods section. In our work, we have shown that bioblending PLA with bio-based PA leads to a more resistant material from a thermal point-of-view.

Round 2

Reviewer 2 Report (New Reviewer)

Some corrections in the manuscript have been made and doubts have been clarified. There is still the question of the advisability of mixing PA and PLA. I understand the author's intentions. How does the research material meet the criteria, The use of such mixtures in the automotive industry raises doubts. PLA is still an expensive material and losing its prime use may raise doubts.

Author Response

As it was commented in our previous response to your kind observation, in applications of long time term service, biodegradability is of secondary importance or even a disadvantage. More emphasis is placed on the content of raw material of renewable origin in order to reduce CO2 emissions associated with the product.

In fact, lately this sector has been interested in including PLA based materials in applications with moderate thermal performance. However, its high brittleness is still a problem. According to the work carried out by the authors, the proposed PLA/BioPA bioblends have managed to capture interest when the PA phase generates a microfibrillated morphology during processing, which is called an "In situ microfibribular composite", of special interest in additive manufacturing processes such as fused filament fabrication (FFF). In studies already published by the authors, under the FFF processing conditions used, they have established that for bioblends with 30% by weight of PA, the resulting morphology after FFF is interconnected fibrillar, promoting an increase (compared to PLA) of 20 times the elongation at break (in unidirectional fill pattern) and up to 7 times in fracture toughness in the interleaved layers printed at 0°/45°/90°/-45° fill pattern. More details of this study can be found in the following reference: 

Martinez-Orozco, N. León, J. Cailloux, M. Sánchez-Soto, M.L. Maspoch, O. Santana, EcoBlends’up: PLA/BioPA blends composites, microfibrillated “in situ” through additive manufacturing, Theoretical and Applied Fracture Mechanics, 118 (2022) 103255. https://doi.org/10.1016/j.tafmec.2022.103255

Despite the price of PLA, Toyota is using PLA for some car interior parts. Because of PLA brittleness, its blending with PA brings improvements.

This manuscript is a resubmission of an earlier submission. The following is a list of the peer review reports and author responses from that submission.

Round 1

Reviewer 1 Report

The paper is very well written (except for some typos and some colloquial sentences), and the work’s topic is of great interest.

However, this work lacks originality as the bioblends described, and many of the conclusions the authors arrive at, are identical to those published by the same authors in a previous paper: Carrasco, F.; Santana Pérez, O.; Maspoch, M.L. Kinetics of the Thermal Degradation of Poly(lactic acid) and Polyamide Bioblends. Polymers 2021, 13, 3996. https:// doi.org/10.3390/polym13223996. However, this article is not reported as a reference in the present paper.

.

The experimental part in the two papers (the present manuscript and the one published in 2021) can be superimposed except for the rheological measurements and the morphological investigation

Overall, the paper looks like a restyling of published data and conclusions already widely described and discussed in the 2021 paper. By example only, Table 2 of this manuscript reports the same data as in the 2021 paper without citing it, and the same applies to Table 4.

the description and discussion of the bioblands new data are reduced to about one page of the entire manuscript. The whole of these considerations makes the manuscript unsuitable for publication.

Author Response

I am enclosing the reply to reviewer 1 comments/questions.

Reviewer 2 Report

I have read in detail the manuscript “Improvement of the thermal stability of polymer bioblends by means of reactive extrusion” submitted to Polymers MDPI.

The subject of this work is interesting, and the authors gathered valuable information on new strategy about the improving thermal stability of PLA/PA blends by chain extension during extrusion. The document is comprehensive, the discussion is reasonable, meanwhile, there are some missing data in the polymer properties and characterization and some corrections need to be made for the current data. So, I think this document should be reconsidered for publication in Polymers only after proper modifications. Some of my specific comments are below:

Point 1: In the abstract “styrene-acrylic multi-functional-epoxide oligomeric reactive agent. Please, indicate that it is the  commercial Joncryl product.

Point 2. Please comment about the possibility for synthesis of the styrene-acrylic multi-functional-epoxide copolymer by controlled polymerization techniques and references about applications of these copolymers as chain extender by PLA.

Point 3:  In the abstract “PLA were responsible for an increase of the onset decomposition temperature”. Please comment about values achieved in their work. Avoid generalizations.

Point 4:  For the sentence “It was shown that reactive extrusion together to higher content of PA resulted in an increased protective effect against … by an increase in activation energy. Avoid generalizations.

Point 5:  For the sentence “The improvement of the thermal stability of bioblends by means of reactive extrusion was explained by an enhanced rheological behavior and morphological considerations. Avoid generalizations, include values that demonstrate the improvement.

Point 6. For the text “In previous works we verified that the best dispersion route was to apply at least two successive passes … changes in the rheological properties, such as a decrease in the viscosity of the melt”. Please include corresponding references.

Point 7. For the text “In order to alleviate the effects of the degradation … such as the increase in shear viscosity and melt-elasticity”. Please include corresponding references.

Point 8. For the sentence: Likewise, it has been confirmed that there are unreacted epoxy groups inserted in the modification that have not taken part in the extension/branching reactions [11]. Additional explication is necessary.

Point 9. In methodology “A commercially extrusion-grade PLA (Ingeo 4032DR and PA10.10”. Other molecular parameters, flow melt index, molecular weight, must be included for a better compression understanding.

Point 10. PLA, as received. Moisture  of the polymers does not affect the chain extension and results obtained in the extrusion process?

Point 11. Please include the name for the corotating twin-screw extruder.

Point 12. For the sentence “This indicates the following reactive extrusion of PLA is able to reduce the number of shorter macromolecular chains which are the first to decompose”. Please comment about the level chain extension and  number of epoxy groups and weight molecular for Joncryl and the effect of these parameters that would be anticipated.

Point 13. Please comment on the trends observed for Tonset  in Figure 1.

Point 14. Please define bioblend conversion.

Point 15. For the Figure 4  indicate symbols and lines meaning.

Author Response

I am enclosing the reply to reviewer 2 comments/questions.

Reviewer 3 Report

The idea of reactive blending PLA has been well published in this field for long run, and the authors provide some additional works to make this concept more complete, which is good to see. Regarding the content, the reviewer has some opinions to make this article more perfect:

1. Please add a picture as Figure 1 and TOC, which provide a big picture of your research work, and help the reader quickly understand your contribution.

2. The introduction part lack of detail insight on blending thermodynamics. Bates (UMN), Lodge (UMN) and Cochran (ISU) have published several PLA modifiers work in recent years, which have detailed thermodynamical insights. Can the authors compare/support the results and former works?

3. SEM image sets need to add neat PLA and neat PA, to compare the differences before/after rex-blend. From my viewpoint, it looks like your sample w/o rex undergo macrophase separation, which make sense, but the scale size looks pretty uniform, which is unusual. Can the author explain this and compare with afford-mention literatures?

4. Please re-draw Figure.2 and 3, the style and ticks are hard to read for most of the readers.

5. Looks like the author didn't explore chemical detail of degradation, which I believe is one of discussion parameter of your research motivation, i.e., the NMR and GPC analysis, if possible, please supply these and make some description; if not possible, please compare the results with some tangible literatures.

6. Please attach DSC raw plots of neat PLA, neat PA, PLA/PA non-rex-blend, and PLA/PA rex-blend in supporting info or main article.

Author Response

I am enclosing the reply to reviewer 3 comments/questions.

Round 2

Reviewer 3 Report

Following reply in the sequence of the original questions proposed by the reviewer.

1. The authors still not answer the suggestions previously provided. The research indeed has some good innovative output, but the presentation style is not attractive. Therefore, I previously suggested a picture to graphically illustrate the entire work.

2. The author still didn't answer this question. The thermodynamics maybe not be part of the scope of this research, but the explanation of how PLA and PA are compatible should be briefly explained by comparing them with the literature.

3. The authors generally reply to the previous questions, which makes more sense.

4. The authors generally reply to the previous questions, which makes more sense.

5. The authors generally reply to the previous questions, which makes more sense.

6. Generally, DSC will be used to support compatibility. Like the Tg positions and other thermal behaviors.
